 

# A phase 2 single center open label randomised control trial for convalescent plasma therapy in patients with severe COVID-19

Yogiraj Ray[1,2✉], Shekhar Ranjan Paul[1], Purbita Bandopadhyay[3,4], Ranit D'Rozario[3,4], Jafar Sarif[3,4], Deblina Raychaudhuri[3], Debaleena Bhowmik[5,4], Abhishake Lahiri[5,4], Janani Srinivasa Vasudevan[6], Ranjeet Maurya [6], Akshay Kanakan[6], Sachin Sharma[6], Manish Kumar[6], Praveen Singh[6], Rammohan Roy [1], Kausik Chaudhury[1], Rajsekhar Maiti[1,7], Saugata Bagchi[1], Ayan Maiti[1], Md. Masoom Perwez[1], Abhinandan Mondal[1], Avinash Tewari[1], Samik Mandal[1], Arpan Roy[1], Moumita Saha[1], Durba Biswas[8], Chikam Maiti[8], Ritwik Bhaduri [9], Sayantan Chakraborty[10], Biswanath Sharma Sarkar[1], Anima Haldar[1], Bibhuti Saha[2], Shantanu Sengupta [6,4], Rajesh Pandey[6,4], Shilpak Chatterjee[3], Prasun Bhattacharya[8], Sandip Paul [5] & Dipyaman Ganguly [3✉]

A single center open label phase 2 randomised control trial (Clinical Trial Registry of India No. CTRI/2020/05/025209) was done to assess clinical and immunological benefits of passive immunization using convalescent plasma therapy. At the Infectious Diseases and Beleghata General Hospital in Kolkata, India, 80 patients hospitalized with severe COVID-19 disease and fulfilling the inclusion criteria (aged more than 18 years, with either mild ARDS having PaO2/FiO2 200–300 or moderate ARDS having PaO2/FiO2 100–200, not on mechanical ventilation) were recruited and randomized into either standard of care (SOC) arm ($N = 40$) or the convalescent plasma therapy (CPT) arm ($N = 40$). Primary outcomes were all-cause mortality by day 30 of enrolment and immunological correlates of response to therapy if any, for which plasma abundance of a large panel of cytokines was quantitated before and after intervention to assess the effect of CPT on the systemic hyper-inflammation encountered in these patients. The secondary outcomes were recovery from ARDS and time taken to negative viral RNA PCR as well as to report any adverse reaction to plasma therapy. Transfused convalescent plasma was characterized in terms of its neutralizing antibody content as well as proteome. The trial was completed and it was found that primary outcome of all-cause mortality was not significantly different among severe COVID-19 patients with ARDS randomized to two treatment arms (Mantel-Haenszel Hazard Ratio 0.6731, 95% confidence interval 0.3010-1.505, with a P value of 0.3424 on Mantel-Cox Log-rank test). No adverse effect was reported with CPT. In severe COVID-19 patients with mild or moderate ARDS no significant clinical benefit was registered in this clinical trial with convalescent plasma therapy in terms of prespecified outcomes.

A full list of author affiliations appears at the end of the paper.

The ongoing pandemic caused by the novel coronavirus SARS-CoV-2 infection has already claimed close to 5.3 million lives, with more than 250 million documented infections worldwide. The acute respiratory disease caused by SARS-CoV-2 infection, the coronavirus disease 2019 or COVID-19, presents with a plethora of symptoms usually found to be spread over two distinct temporal phases in patients who are symptomatic. The symptoms in the initial milder phase variably include malaise, fatigue, fever, cough, loss of smell and taste and diarrhoea in some, mostly followed by recovery[1]. But in a considerable fraction of patients, this milder phase, later on, culminates in more severe disease, characterized by gradually worsening hypoxemia requiring exogenous $O_2$ supplementation. In some patients with this severe disease, a progression to acute respiratory distress syndrome (ARDS) is encountered, leading to untoward fatal outcomes in a number of them[1,2]. Individuals having metabolic co-morbidities have been shown to have a predilection for COVID-19 disease severity[3]. An aberrant hyperactivation of the immune system has been found to be associated with these severe symptoms, most notably characterized by a systemic deluge of inflammatory cytokines or 'cytokine storm'[4–6].

Apart from the medical interventions aimed at mitigating symptomatologies, different therapeutic approaches are currently being explored, either by repurposing specific anti-viral agents, viz. remdesivir[7], or by using corticosteroids to affect immunomodulation[8], to treat patients progressing to severe disease. A number of patients also present with intravascular thrombosis and hence a role for prophylactic and therapeutic anticoagulation has also found a place in the standard of care in severe patients[9]. But in the absence of proven efficacy of any specific pathogen-targeted therapy, convalescent plasma (CP) transfusion is an age-old strategy for passive immunization, with the primary intention to supplement non-recovering patients with antibodies against specific pathogens[10]. Convalescent plasma therapy (CPT) has emerged as a widely tried strategy against COVID-19 too, having been explored in a large number of clinical trials all over the world[11–37]. Results of a multitude of randomized control clinical trials and efforts at meta-analysis revealed scarce evidence for significant clinical benefits of convalescent plasma therapy in COVID-19[12–21,32], while others reported contradictory data[22–27,33,34].

We report here insights gathered from a single-center open-label phase II randomized control trial done in Eastern India, on patients with severe COVID-19 disease with evidence for progressing to mild to moderate acute respiratory distress syndrome and identify the clinical and immunological benefits of CP transfusion. The trial was registered with the Clinical Trial Registry of India (No. CTRI/2020/05/025209). The primary outcomes were all-cause mortality on day 30 after enrolment and identification of immunological correlates of response to CPT, if any. The pre-specified secondary outcomes were time to recovery from ARDS, time taken to register negative RT-PCR and documenting any adverse effects on receiving CPT. In this study we find no significant clinical benefit in patients receiving CPT in terms of either survival benefit or reduction in the duration of hospital stay. While addressing the primary outcome of immunological correlates of convalescent plasma therapy we characterize a potential anti-inflammatory role of CP.

## Results

**Recruitment of convalescent donors and characterization of antibody response.** Convalescent individuals ($N = 61$, female: $N = 12$, Age: $26 \pm 2.98$ years; male: $N = 49$, Age: $35.37 \pm 9.06$ years) who recovered from COVID-19 at least 28 days prior to screening, were screened for eligibility for plasma donation.

Forty-six donors were found eligible for plasmapheresis. All of them were screened at 40–80 days after they were first tested positive on RT-PCR for SARS-CoV-2 (Supplemental Fig. 1A). The nature of their disease course was assessed to be between 1 and 5 on the WHO Clinical Progression score with the majority having suffered from the mild symptomatic disease.

On measuring anti-SARS-CoV-2 spike protein IgG content of CP a significant correlation was found with the age of donors—with increasing age a more robust humoral response and higher IgG content was noted (Fig. 1A). Interestingly in this small cohort of convalescent donors there was no significant correlation between time passed since positive RT-PCR test and SARS-CoV-specific IgG content of CP (Supplemental Fig. 1B). Specific IgG content was also not correlated with WHO clinical progression scores for the disease course reported by the donors (Supplemental Fig. 1C). Notably a very strong correlation was noted between anti-SARS-CoV-2 spike IgG content of CP and the content of neutralizing antibody (nAb) (assessed by a surrogate in vitro assay[38], measuring the ability of CP to block the interaction between SARS-CoV-2 spike receptor-binding domain

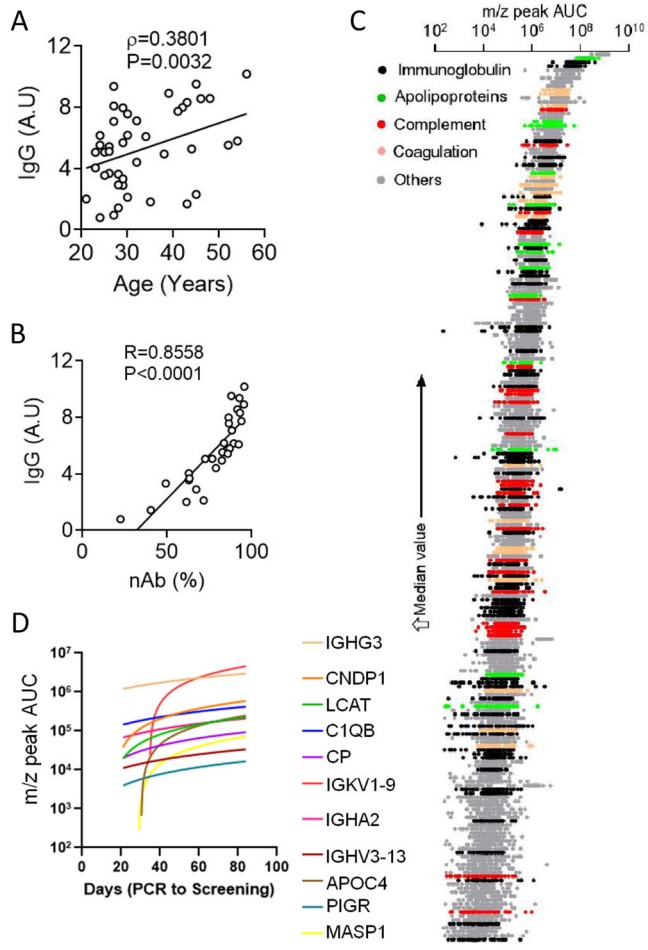

**Fig. 1 Convalescent donor characteristics. A** Correlation between the anti-SARS-CoV-2 spike IgG content of CP and age of donors. **B** Correlation between the anti-spike IgG content of CP and the efficiency to neutralize spike protein RBD-ACE2 interaction. **C** A scatter plot showing the relative abundance of the constituents of CP proteome with colour coding for major functional families. **D** Statistically significant correlation (Pearson correlation, $p$-value < 0.05, one tailed) of CP constituents with the time since RT-PCR positivity of the donors. Spearmann or Pearson correlation was computed and the corresponding $\rho$ or R and two-tailed $P$-values are noted when significant correlation is found.

and its cognate receptor angiotensin-converting enzyme 2(*ACE2*) (Fig. 1B). Proteomic analysis of CP was also performed using mass spectrometry for characterization of the major protein components of plasma. As expected a large number of immunoglobulin subtypes were featured in the proteomic profile of CP (Fig. 1C). The proteome content of CP was also featured by a number of complement system-related proteins, proteins relating to the coagulation pathways, apolipoproteins and a number of anti-inflammatory proteins, in addition to the immunoglobulins as expected (Fig. 1C, Supplemental Table 1). A number of immunoglobulins showed a significant positive correlation with time passed since the diagnostic RT-PCR, as did a number of other proteins as well (Fig. 1C). More interestingly, specific immunoglobulin components (viz. immunoglobulin kappa variable 1–9) showed a late-onset rapid accumulation (Fig. 1D), which warrants future exploration of the antibody response in deeper detail.

**Inter-arm patient characteristics with viral load and humoral response.** We recruited 80 patients, fulfilling the inclusion criteria (aged more than 18 years, admitted with severe COVID-19 disease having either mild ARDS having $PaO_2/FiO_2$ 200–300 or moderate ARDS having $PaO_2/FiO_2$ 100–200, not on mechanical ventilation and within 5–10 days from initial presentation), into the trial and randomized into either standard of care (SOC, $N = 40$) arm or the convalescent plasma therapy (CPT, $N = 40$) arm (Fig. 2). Demographic characteristics of the patients between two parallel arms were not significantly different (Supplemental Table 2, Supplemental Fig. 2A). All patients recruited in CPT arm received two transfusions of 200 ml ABO-matched CP on two successive days, first one being on the day of enrolment, except for one patient who succumbed before he could be transfused with the second unit, but not due to any adverse effect that could be temporally linked to the first CP transfusion. Transfusion-related adverse effects were reported in none of the patients in

CPT arm. All the patients recruited had received similar standards of care (Supplemental Table 2).

Viral loads at the day of enrolment (T1) were comparable between patients in the two arms (Supplemental Fig. 2B). Viral isolates could be sequenced from nasopharyngeal swabs collected from 25 patients in the SOC arm and 27 patients in the CPT arm. There was no significant difference in viral clade representations between the two arms. Six patients in the SOC arm and 13 patients in the CPT arm showed infection with SARS-CoV-2 clade 19 A, 18 in SOC arm and 13 in CPT arm with 20 A and one in each arm with the clade 20B (Fig. 3A). The sequenced viral isolates were collected from recruited patients all of whom came from the city of Kolkata in the state of West Bengal in the eastern region of India. On analyzing the SARS-CoV-2 genome sequences submitted in GISAID from this region of India we found that viral clade representations among our patients corroborated with the contemporaneous clade-prevalence in the city (see details in online methods and Supplemental Table 3).

Neutralizing antibody content of plasma at T1 was also not significantly different between SOC and CPT (Supplemental Fig. 2C). A significant correlation between neutralizing antibody content of plasma was noted with the number of days passed since the patients first got tested positive for SARS-CoV-2 on RT-PCR (Fig. 3B). Across all patients, from both arms, a significant negative correlation between neutralizing antibody content of plasma at T1 and concomitant viral load was found, as expected (Fig. 3C). A great majority of recruited patients randomized into either group had documented co-morbidities, viz. type 2 diabetes, hypertension, coronary heart disease, hypothyroidism and others, but the relative representations of these co-morbidities were comparable between two arms (Fig. 3D, Supplemental Table 2). Data on the routine clinical investigations were also found to be comparable among the groups (Supplemental Table 2).

**Comparison of primary and secondary outcomes between trial arms.** On analyzing the primary outcome of all-cause mortality at 30 days we registered no significant difference in survival between the two arms (Fig. 4A, Mantel-Haenszel Hazard Ratio 0.6731, 95% confidence interval 0.3010–1.505, with a *P*-value of 0.3424 on Mantel-Cox Log-rank test).

Another primary outcome of our trial had been identifying immune correlates of response to therapy, if any. Severe COVID-19 patients have been found by previous studies to experience a systemic hyper-inflammation characterized by a cytokine deluge. We have previously characterized the nature and dimension of this so-called cytokine storm in a fraction of these patients, comparing them to patients with mild COVID-19 disease[39]. On measuring plasma abundance of a panel of 48 cytokines in patients from both arms we found that correlative nature and magnitude of the individual components of the cytokine storm were not notably different at T1 in correlative network analysis (Fig. 4B, C). Data from a panel of 36 cytokines were included in all analyses, selected based on their measurable plasma abundance in at least 70% of the patients. The magnitude of plasma abundance was computed in comparison with the median abundance of individual cytokines in patients having mild COVID-19 disease reported as earlier[39]. Quite similar to this earlier study, a more significant attenuation of the systemic deluge of cytokines at T2 was noted in patients in the CPT arm, in terms of calming down the correlative upregulation (Fig. 4B, C). This was also evident from the reduction in the median abundance of major pathogenically significant cytokines as well as in terms of the number of patients in CPT arm registering such a change (Supplemental Fig. 3).

The secondary outcome of time taken for recovery from ARDS in all patients could not be determined accurately for all patients

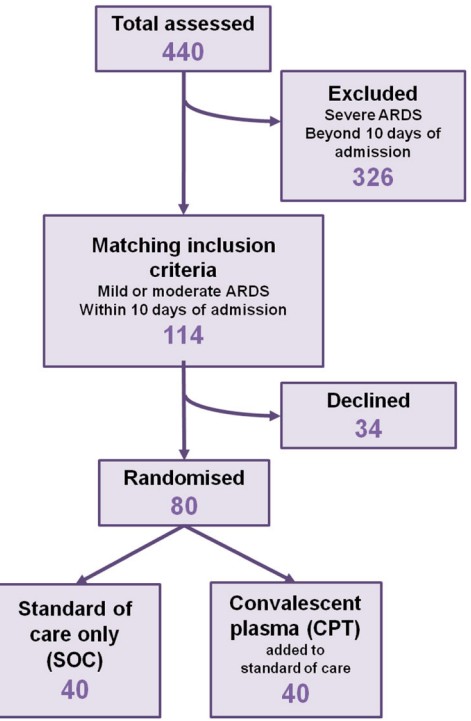

**Fig. 2 Clinical trial design and patient recruitment.** Diagram representing the design of the randomized control trial.

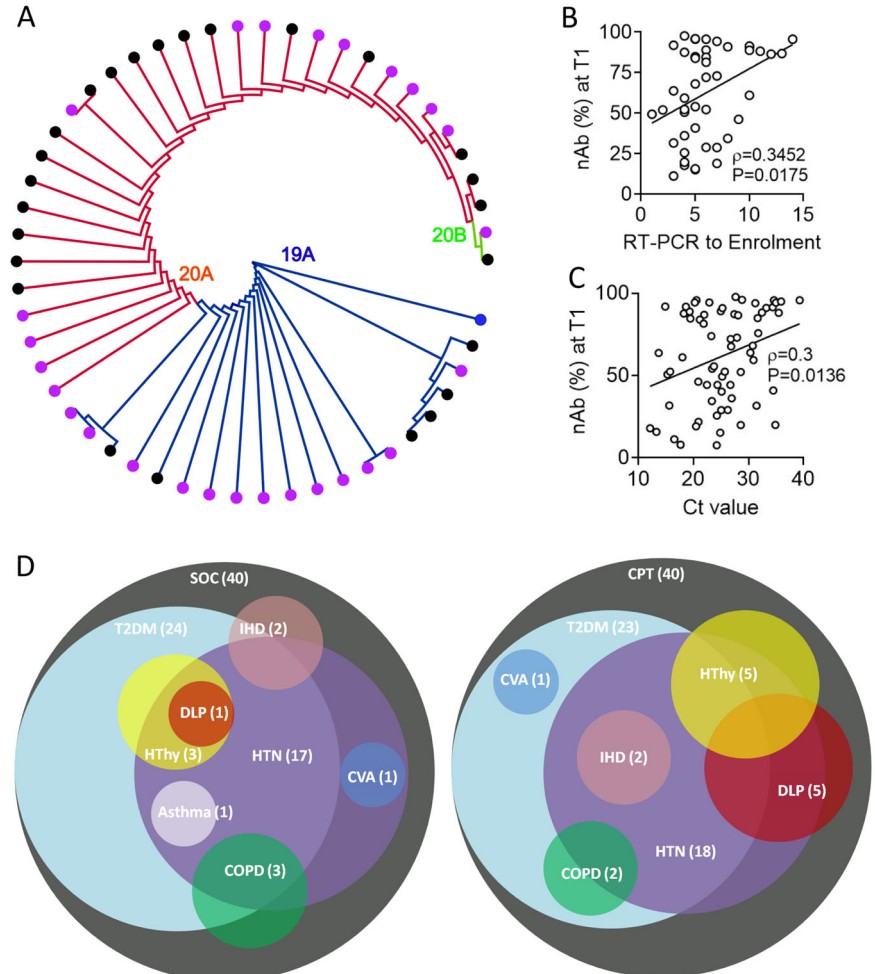

**Fig. 3 Key pathogen and host characteristics and primary clinical outcomes. A** The whole-genome phylogenetic tree of SARS-CoV-2 viral isolates from SOC and CPT groups. Black and purple tips represent the samples from SOC and CPT groups. **B** Correlation between the neutralizing antibody content of plasma at T1 and day since the patients were tested positive on RT-PCR. **C** Correlation between the neutralizing antibody content of plasma at T1 and CT values obtained from SARS-CoV-2 RT-PCR from nasopharyngeal swabs collected at T1 is shown. **D** Venn diagram showing distribution of different co-morbid conditions among the patients in SOC and CPT arms.

due to emergent operational limitations in access to computed tomography facility and arterial blood gas analysis for follow-up. Instead, recovery from COVID-19 disease was assessed in terms of time taken for discharge from the hospital, although it was not pre-specified in the trial protocol. Across all patients, we found no significant benefit in the CPT arm, either in terms of duration of hospital stay since the day of enrolment (Fig. 5A, median of 17 days for SOC vs 13 days for CPT arm, *P*-value of 0.098 on Mantel-Cox Log-rank test) or duration of hospital stay since admission (Fig. 5B, the median of 23 days for SOC vs 17 days for CPT arm, *P*-value of 0.0797 on Mantel-Cox Log-rank test). Disease course records for individual patients are given in Supplemental Table 4. Among the other secondary outcomes, a comparison of time taken for the patients in the two arms to register negative RT-PCR for SARS-CoV-2 could not be done due to statutory suspension of clinical use of repeat RT-PCR among hospitalized patients, which was to be the source of this data. Finally, no transfusion-related adverse effects were documented in any of the patients in the CPT arm.

Most of the patients recruited had moderate ARDS at the time of recruitment with a mean $SpO_2$ ($O_2$ saturation in the capillary blood)/ $FiO_2$ (Fraction of $O_2$ inhaled) ratio of 108.38 on the day of enrolment for SOC arm and 111.43 for the CPT arm (Supplemental Fig. 2D). Daily recorded respiratory rates over

the course of hospitalization from the day before enrolment was found to be comparable between two arms, as were basic parameters of hemodynamic regulation, viz. heart rate, systolic and diastolic blood pressures (Supplemental Fig. 2E-H). Figure 5C shows the kinetics of $SpO_2/FiO_2$ ratio (S/F ratio) on a colour scale plotted daily since the day of admission till discharge/death for all patients recruited in either arm (Fig. 5C). Compared between two arms, we found no significant relative benefit in terms of mitigation of hypoxia as represented by the kinetics of S/F ratio over 10 days following enrolment (Fig. 5D), which was done as a non-pre-specified exploratory analysis.

## Discussion

The open-label randomized control trial for passive immunization of severe COVID-19 patients with CPT adds to the growing literature on similar trials of different designs and sample sizes. The present RCT was done in a low clinical resource setting in a single center. The clinical outcome comparisons did not reveal a significant relative benefit on receiving convalescent plasma therapy in severe COVID-19 patients, most of who had progressed to moderate acute respiratory syndrome.

A large number of clinical trials, both randomized control and matched-control ones, have been ongoing in different parts

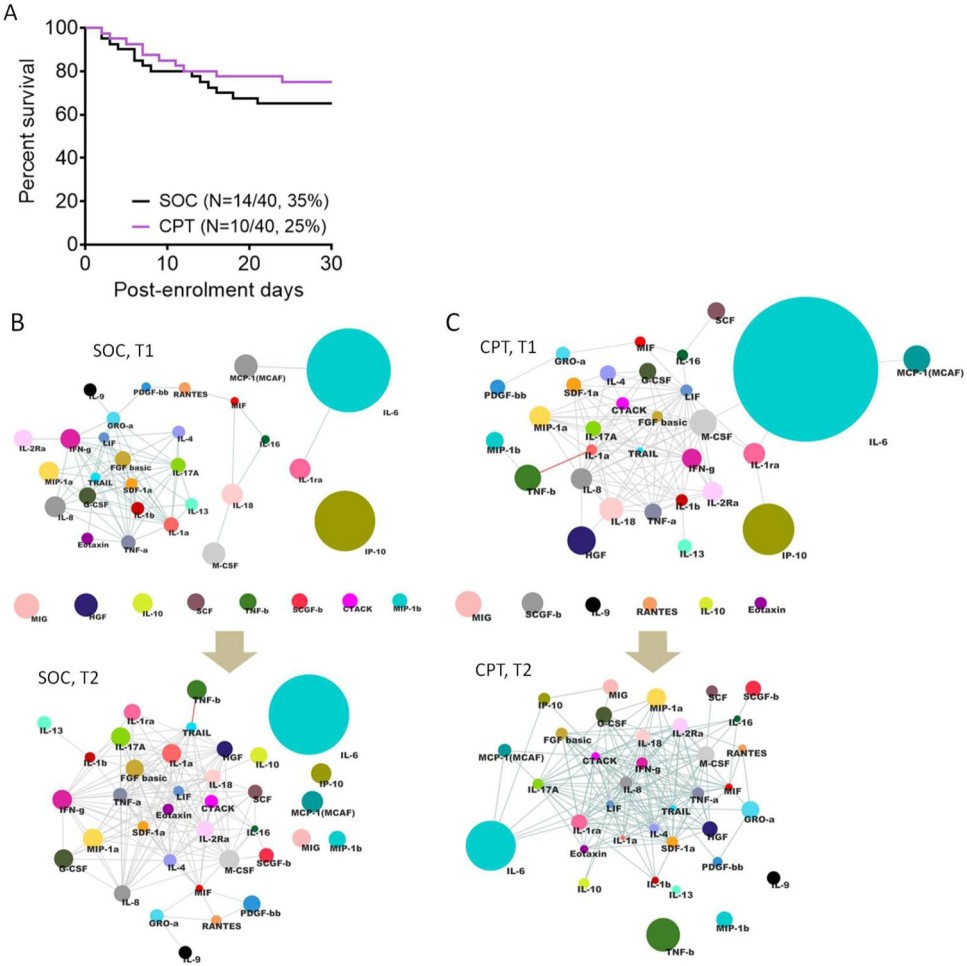

**Fig. 4 Primary outcomes compared between two arms of the trial. A** Survival of patients in the two arms from the day of enrolment till day 30 post-enrolment are compared in a Kaplan–Meier curve, for all age groups. Surviving patients were censored on day 30 post-enrolment. For all outcomes Mantel-Cox log-rank test was performed and corresponding *P*-values are only shown for statistically significant differences. Panels **B** and **C** describe the correlation network for 36 cytokines measured in plasma of severe COVID-19 patients, from both SOC and CPT group at T1 and T2 timepoints, respectively. The diameter of the nodes represents extent of enriched abundance compared against a median value derived from patients with mild disease. Edges are shown only for Spearman correlation value of 0.7 and above.

of the world since very early into the SARS-CoV-2 pandemic. They varied in study designs, sample sizes and scopes, more importantly even in the registered outcomes. A large trial in USA established the safety of this strategy of passive immunization[11]. But a number of RCTs reported no significant clinical benefit in the severe COVID-19 patients receiving convalescent plasma[12–21]. On the other hand, contradictory reports of some clinical benefits also have been there, both from matched-control studies[22–27] as well as a few RCTs[28–31]. Different meta-analytic efforts also reported data showing both efficacy and inefficacy of CP in COVID-19[32–37].

An important revelation of this trial has been the prominent anti-inflammatory effect of CPT, in terms of more prominent attenuation of the systemic surge of a large panel of cytokines compared to the standard care, perhaps due to CP proteome consisting of a number of anti-inflammatory proteins. The biology underlying the lack of response to convalescent plasma therapy in severe COVID-19 patients, despite this discernible anti-inflammatory effect, will be of great interest in subsequent mechanistic studies as well as in the context of therapeutic usage of specific monoclonal antibodies in COVID-19.

The major limitation of the present trial had been a small sample size, which also perhaps prevented the trial from

discerning the relative clinical benefits. Moreover, the trial was open-label and the allocation of therapies was not concealed following randomization, which is another limitation of this trial. Altogether, this randomized control trial showed no relative clinical benefit in response to convalescent plasma therapy in severe COVID-19 patients as per the pre-specified primary outcome.

## Methods

**Ethical approval**. The randomized control trial (RCT) on passive immunization with convalescent plasma therapy and all associated studies were done with written informed consent from the patients according to the recommendations and ethical approval from the Institutional Review Boards of all the concerned institutions, viz. CSIR-Indian Institute of Chemical Biology, Kolkata, India (IICB/IRB/2020/3 P), Medical College Hospital, Kolkata (MC/KOL/IEC/NON-SPON/710/04/2020), India and Infectious Disease & Beleghata General Hospital (ID & BG Hospital), Kolkata, India (IDBGH/Ethics/2429). The RCT was approved by Central Drugs Standard Control Organisation (CDSCO) under Directorate General of Health Services, Ministry of Health & Family Welfare, Govt. of India (approval no. CT/BP/09/2020) and registered with Clinical Trial Registry of India (CTRI, No. CTRI/2020/05/025209), under Indian Council of Medical Research, India. The study was conducted in accordance with the Declaration of Helsinki.

**Collection of convalescent plasma**. Convalescent donors were recruited and screened at the Department of Immunohematology and Blood Transfusion,

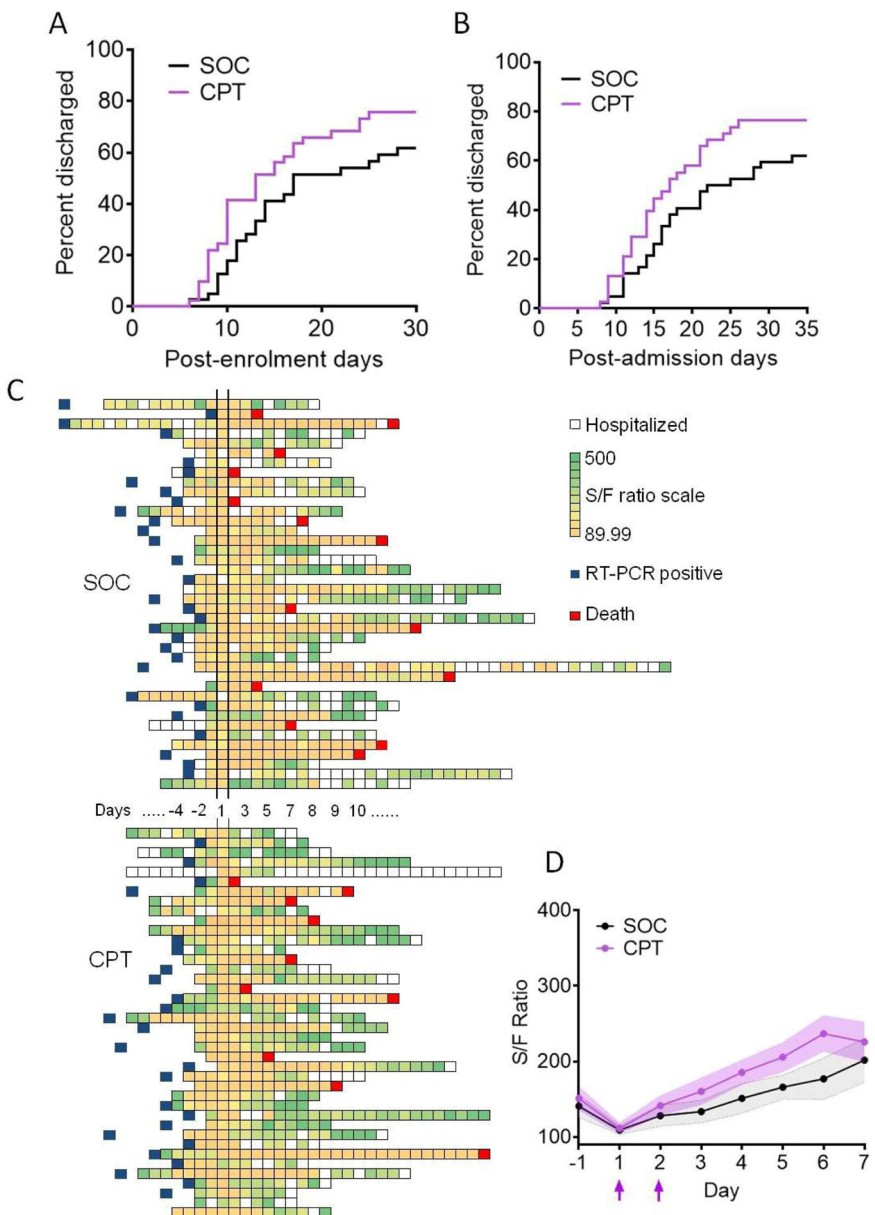

**Fig. 5 Secondary and exploratory clinical outcomes compared between two arms. A** Total hospital stay duration of the patients from both arms are plotted in an ascending Kaplan–Meier curve, for all age groups. Deaths and non-remission at day 35 post-admission were censored. **B** Hospital stay duration of the patients from both arms since the day of enrolment are plotted in an ascending Kaplan–Meier curve, for all age groups. Deaths and non-remission at day 30 post-enrolment were censored. **C** Representative plot of $SpO_2/FiO_2$ kinetics (shaded as per the colour scale depicted) of individual patients during hospitalization. The blue box denotes day of diagnosis through RT-PCR for SARS-CoV-2 and the red boxes denote time of death. **D** The ratio between saturation of O2 in blood ($SpO_2$) and fraction of O2 received (FiO2), or S/F ratio is plotted for patients in SOC (black line) and CPT (purple line) arms from the day before enrolment till the 7th day post-enrolment, among all age groups. Purple arrows indicate the days when convalescent plasma was transfused. 95% confidence interval is shown for each group.

Medical College Hospital, Kolkata, India. The inclusion criteria for donors were: age >18 years, males or nulliparous female convalescent volunteers with a history of being positive for SARS-CoV-2 on RT-PCR, having weight >55Kg, complete resolution of symptoms at least 28 days prior to donation, and a negative RT-PCR test for SARS-CoV-2 before plasma donation. Consenting convalescent patients not fit to donate blood based on the history and examination, who have had a transfusion of blood products in last year were excluded from donation. A questionnaire was used to collect data on the disease course from all convalescent donors. On the screening day, peripheral blood samples were drawn for the following pre-donation tests: blood group (ABO grouping and Rh phenotyping) and antibody screening for clinically significant antibodies (Extended Rh, Kell, Duffy, Kidd, MNS, antibody screen positive donors were excluded), complete blood count including hemoglobin, hematocrit, platelet count, total and differential leucocyte count (Hb > 12.5 g/dl, platelet count > 150,000 per microliter of blood and TLC within normal limits were included), screening for HIV, HBV and HCV, MP and

syphilis by serology and ID-NAT for Hep B and C and HIV1 (all non-reactive donors by both tests were included), total serum protein (donors with total serum protein >6 gm/dl will be accepted, as per Drugs and Cosmetics (Second Amendment) Rules, 2020. For the initial 18 convalescent donors pre-donation screening for anti-SARS-CoV-2 spike protein IgG content of their plasma could not be done due to the absence of dependable assay kits. Once it was available (Euroimmun) all donors were also pre-screened for anti-SARS-CoV-2 spike IgG. For the first 18 donors it was done retrospectively. All donated plasma were tested for their neutralizing antibody content using an in vitro surrogate neutralization kit[18]. For pre-screened donors a value of 1.5 for the ratio optical density between the sample and calibrator was taken as a cut-off for inclusion. A fraction of the convalescent plasma sample was also characterized for their proteome using LC-MS/MS (described below). Plasmapheresis on eligible donors was done on a Haemonetics MCS + Cell Separator. Four hundred millilitre of plasma was collected and aliquoted with sterile connections (Terumo TSCD) in two plasma bags containing

200 ml each and cryostored at −80 °C, until commissioned for transfusion in an ABO-matched recipient.

**Trial design**. The inclusion criteria for recruitment of severe COVID-19 patients as recipients of convalescent plasma in this open-label phase II randomized control trial were: consenting patients admitted with RT-PCR proven COVID-19 with severe disease (fever or suspected respiratory infection, plus one of the following; respiratory rate >30 breaths/min, severe respiratory distress, SpO$_2$ < 90% at room air) with mild ARDS, defined as patients having a partial pressure of oxygen in the arterial blood (PaO$_2$)/fraction of inspired oxygen (FiO$_2$) ratio of 200–300 mmHg or moderate ARDS, defined as PaO$_2$/FiO$_2$ 100–200 mmHg, not on mechanical ventilation. Pregnant or breastfeeding mothers, patients with age less than 18 years, patients participating in any other clinical trial, patients having any clinical condition precluding infusion of blood products were excluded. The trial was designed to recruit 40 patients in each arm, based on sample size calculation based incidence of the specified primary outcome among patients presenting with mild and moderate ARDS and receiving standard of care (as per guidelines of Indian Council of Medical Research) for the preceding one month in the chosen clinical trial site (28% with standard of care, with hypothesized incidence of the same in response to CPT taken to be 5%). Accordingly, 80 patients were enrolled in the trial at ID & BG Hospital, Kolkata, India, the first patient being recruited on May 31, 2020 and the last on October 12, 2020. A digitally derived random sequence divided into four blocks of twenty was used for randomizing the patients. Patients fulfilling the inclusion criteria were randomized into either the SOC arm receiving only the standard of care therapy, as per current advisory, or into the CPT group to receive two consecutive doses of ABO-matched 200 ml convalescent plasma on two consecutive days, the first transfusion being on the day of enrolment, in addition to standard of care. A computer-generated random sequence, communicated to the trial site on the day of enrolment as each patient meeting the inclusion criteria were identified. After randomization of the patients the allocations were not concealed. The primary outcomes were all-cause mortality on day 30 after enrolment and identification of immunological correlates of response to CPT, if any. The pre-specified secondary outcomes were time to recovery from ARDS (time till the day of discharge from hospital), time taken to register negative RT-PCR and documenting any adverse effects on receiving CPT. Nasopharyngeal swabs, fecal samples and peripheral blood in EDTA vials were collected on the day of enrollment (time point 1 or T1). Then on the 3rd day or 4th day post-enrollment (time point 2 or T2) and finally on the 7th day post-enrollment (time point 3 or T3) peripheral blood samples were taken.

**Standard-of-care**. At the clinical trial site (ID & BG Hospital, Kolkata, India) standard-of-care (SOC) in all patients with evidence for ARDS were: O$_2$ therapy as per requirement, either intravenous or oral corticosteroids, for patients with D-dimer <1000 Fibrinogen Equivalent Units (FEU) prophylactic anticoagulation and for patients with D-dimer >1000 ng/ml FEU therapeutic anticoagulation using either low molecular weight heparin or unfractionated heparin, appropriate broad-spectrum antibiotic therapy based on clinical, biochemical and microbiological assessment, appropriate anti-diabetic therapy to maintain blood sugar below 200 mg/dl, anti-hypertensive agents, as per requirement, were used to maintain systolic blood pressure 100-140 mm of Hg, diastolic blood pressure at 70–90 mm of Hg and mean arterial pressure >65 mm of Hg. Awake proning for 6–8 h/day was attempted in all patients with evidence for ARDS. O$_2$ therapy was designed to maintain SpO$_2$ > 95% with the successive deployment of higher efficiency devices as required, viz. nasal cannula, face mask, face mask with reservoir. Patients unable to maintain O$_2$ saturation in the blood (SpO$_2$) above 90% with face mask with reservoir, high flow nasal cannula (HFNC) or in some cases mechanical ventilation (MV) were deployed. For the purpose of kinetic analysis of the SpO$_2$/FiO$_2$ ratio, a value of 89.99 was used for data points where either HFNC or MV was in use. One patient in the SOC arm received Tocilizumab, none in the CPT arm. 13 patients in the SOC arm and 11 patients in the CPT arm received Remdesivir.

**Plasma cytokine analysis**. Plasma was isolated from peripheral blood of patients collected in EDTA vials. Plasma cytokine levels (pg/ml) were measured using the Bio-Plex Pro Human Cytokine Screening Panel 48-Plex Assay (Bio-Rad, Cat No. 12007283, for FGF basic, Eotaxin, G-CSF, GM-CSF, IFN-γ, IL-1β, IL-1RA, IL-1α, IL-2RA, IL-3, IL-12p40, IL-16, IL-2, IL-4, IL-5, IL-6, IL-7, IL-8, IL-9, GRO-α, HGF, IFN-α2, LIF, MCP-3, IL-10, IL-12p70, IL-13, IL-15, IL-17A, IP-10, MCP-1, MIG, β-NGF, SCF, SCGF-β, SDF-1α, MIP-1α, MIP-1β, PDGF-BB, RANTES, TNF-α, VEGF, CTACK, MIF, TRAIL, IL-18, M-CSF and TNF-β), using manufacturer's protocol.

**RNA Isolation from nasopharyngeal swab samples and RT-PCR**. RNA from COVID-19 samples in TRIzol samples were extracted using chloroform-isopropanol method. qRT-PCR for SARS-CoV-2 detection was performed using the STANDARD M nCoV Real-Time Detection kit (Cat No. 11NCO10, SD Biosensor), approved by Indian Council of Medical Research (ICMR), as per the manufacturer's protocol. The RT-PCR was run on QuantStudio 6 Flex Real-Time PCR Systems (Applied Biosystems, Thermo Fisher Scientific). The kit suggested using the cut-off of Ct value 36 for the SARS-CoV-2 genes (*RdRp* and *E* gene) and

the performance of the human positive control gene to declare a sample as SARS-CoV-2 positive. CY5 labeled Internal Control is used as a positive control. Sequencing was attempted for all samples within the said cut-off. Of the 52 sequenced samples, there were 17 samples with Ct > 30, 5 samples with Ct > 35 and the rest 30 samples of Ct < 30.

**SARS-CoV-2 whole-genome sequencing using nanopore platform**. In brief, 100 ng total RNA was used for double-stranded cDNA synthesis by using Superscript IV (ThermoFisher Scientific, Cat. No. 18091050) for first-strand cDNA synthesis followed by RNase H digestion of ssRNA and second-strand synthesis by DNA polymerase-I large (Klenow) fragment (New England Biolabs, Cat. No. M0210S). Double-stranded cDNA thus obtained was purified using AMPure XP beads (Beckman Coulter, Cat. No. A63881). SARS-CoV-2 genome was then amplified from 100 ng of the purified cDNA following the ARTIC V3 primer protocol. Sequencing library preparation consisting of End Repair/dA tailing, Native Barcode Ligation, and Adapter Ligation was performed with 200 ng of the multiplexed PCR amplicons according to Oxford Nanopore Technology (ONT) library preparation protocol-PCR tiling of COVID-19 virus (Version: PTC_9096_v109revE_06Feb2020). Sequencing in sets of 24 barcoded samples was performed on MinION Mk1B platform by ONT.

In terms of sequencing performance, inclusive of genome coverage and sequencing depth, we did not observe a correlation with the Ct values. We found that samples even with high Ct value performed well upon sequencing whereas lower Ct value samples had sub-optimal sequencing output. During analysis, the read quality has been kept uniform for all the samples used in the study irrespective of the background Ct value. Using Nanopore sequencing adapting ARTIC protocol, we have achieved success with higher Ct value samples.

**Nanopore analysis**. The ARTIC end to end pipeline was used for the analysis of MinION raw fast5 files up to the variant calling. Raw fast5 files of samples were basecalled and demultiplexed using Guppy basecaller that uses the basecalling algorithms of Oxford Nanopore Technologies (https://community.nanoporetech.com) with phred quality cut-off score >7 on GPU-linux accelerated computing machine. Reads having phred quality score less than 7 were discarded to filter the low-quality reads. The resulting demultilexed fastq were normalized by read length of 300-500 (approximate size of amplicons) for further downstream analysis and aligned to the SARS-CoV-2 reference (MN908947.3) using the aligner Minimap2 v2.17[40]. Nanopolish was used to index raw fast5 files for variant calling from the minimap output files[41]. To create consensus fasta, bcftools v1.8 was used over normalized minimap2 output.

**Phylogenetic reconstruction**. The assembled SARS-CoV-2 genomes were aligned using MUSCLE aligner in default mode using the software UGENE v34[42,43]. The phylogenetic tree construction was performed using the Maximum Likelihood method. Visualization and further editing of the tree were done in FigTree 1.4.4 (http://tree.bio.ed.ac.uk/software/figtree). Clade Nomenclature of Nextstrain was used to visualize the phylogenetic tree[44]. The SARS-CoV-2 positive samples that could be sequenced were collected from patients who came from the city of Kolkata in the state of West Bengal, India. To corroborate the relatively lower percentage of 20B clades, we analyzed the SARS-CoV-2 genome sequences submitted in GISAID from this region of India. We looked for the GISAID deposited sequences from Kolkata as well as West Bengal (overall) and found 331 and 1065 genome sequences, respectively. The relative percentage of 20 A and 20B clade showed that the 20 A clade was present in higher frequencies compared to 20B. Our analysis from all the available sequences (331 samples) in GISAID from the region of study (Kolkata) till date (10th April, 2021), 60.42% of these sequences are of Nextstrain clade 20 A, while clade 20B is observed at a frequency of 26.59% (Supplemental Table 3). The lower percentage of clade 20B is also reflected when we analyze the total GISAID data from the state of West Bengal, India. Of the total 1065 sequences, clade 20 A contributed 56.62% whereas clade 20B being 33.52% (Supplemental Table 3). We also specifically looked for the month of September–October 2020 (21 in number) from Kolkata, which corroborates the same pattern with clade 20B being 19.05% (Supplemental Table 3). We also double-checked for the base calls at positions G28881A, G28882A and G28883C and found the same pattern for clade 20 A and 20B in the samples sequenced.

**SARS-CoV-2 surrogate virus neutralization assay**. Neutralizing antibodies against SARS-CoV-2 in human plasma samples from peripheral blood of convalescent donors were detected using GeneScript SARS-CoV-2 Surrogate Virus Neutralization kit (Cat no-L00847). The assay was performed according to the manufacturer's protocol. Plasma samples and provided positive and negative controls were diluted at a ratio of 1:10 with the sample dilution buffer. The presence of SARS-CoV-2 neutralizing antibodies in the plasma samples results in inhibition of the interaction between HRP-RBD and plate-bound human ACE2 protein, and subsequent development of colour, assay results are interpreted as inhibition rate of assay reaction. The neutralizing antibody content was measured for all convalescent plasma samples as well as for recipients at different timepoints.

**ELISA for anti-SARS-CoV-2 IgG**. Levels of Immunoglobulin G (IgG) specific for SARS-CoV-2 in the plasma isolated from peripheral blood of convalescent donors were detected using EUROIMMUN Anti-SARS-CoV-2 (IgG) Elisa kit (Cat No- EI 2606-9601 G). This assay provides a semiquantitative estimation of IgG levels against SARS-CoV-2 spike protein. The assay was performed according to the manufacturer's protocol. The presence of anti-SARS-CoV-2 IgG antibodies in the plasma was measured using the following formula: Ratio = Extinction of the control or patient samples/Extinction of calibrator (Ratio ≥ 1.1 is interpreted as positive).

**Proteomics analysis of convalescent plasma**. From each sample, 10 μl of plasma was taken in a fresh 1.5 ml micro-centrifuge tube and diluted to 100 μl with phosphate buffer (1× PBS). Rapid protein precipitation was performed for these samples by addition of 400 μl of acetone and incubated at room temperature for 2 min followed by centrifugation at $10,000 \times g$ for 5 min[45]. After removal of supernatant, pellets were air-dried and resuspended in 100 mM Tris-HCl buffer (pH 8.5). Protein estimation was performed for each samples using the Bradford assay (Sigma–Aldrich, USA). For proteomics analysis, 20 μg of protein from each sample was reduced by the addition of 25 mM of dithiothreitol (Sigma–Aldrich, USA) and incubated at 56 °C for 25 min. Cysteine alkylation was performed by addition of 55 mM iodoacetamide (Sigma–Aldrich, USA) and incubated in dark for 20 min. Samples were subjected to trypsin (sequencing grade, Promega) digestion at an enzyme to substrate ratio of 1:10 for 18 h at 37 °C. The reaction was terminated by the addition of 0.1% formic acid and dried under vacuum. Peptide clean-up was performed using Oasis HLB 1cc Vac cartridge (Waters). DIA-SWATH analysis for samples was performed on a quadrupole-TOF hybrid mass spectrometer (TripleTOF 6600, SCIEX, USA) coupled to a nano-LC system (Eksigent Nano-LC-425). For each sample, 4 μg of these peptides were loaded on a trap-column(ChromXP C18CL 5 μm 120 Å, Eksigent) where desalting was performed using 0.1% formic acid in water with a flow rate of 10 μl per minutes for 10 min. Peptides were then separated on a reverse-phase C18 analytical column (ChromXP C18, 3 μm 120 Å, Eksigent) in a 57 min gradient of buffer A (0.1% formic acid in water) and buffer B (0.1% formic in acetonitrile) at a flow rate of 5 μl/min. Buffer B was slowly increased from 3% at 0 min to 25% in 38 min, further increased to 32% in next 5 min and ramped to 80% buffer B in next 2 min. In 0.5 min, buffer B was increased to 90% and the column was washed for 2.5 min, buffer B was brought to an initial 3% in the next 1 min and the column was reconditioned for the next 8 min. A method with 100 precursor isolation windows was defined based on precursor $m/z$ frequencies using the SWATH Variable Window Calculator (SCIEX), with a minimum window of 5 $m/z$. Data were acquired using Analyst TF 1.7.1 Software (SCIEX), the accumulation time was set to 250 msec for the MS scan (400–1250 $m/z$) and 25 msec for the MS/MS scans (100–1500 $m/z$). Rolling collision energies were applied for each window based on the $m/z$ range of each SWATH and a charge 2+ ion, with a collision energy spread of five. The total cycle time was 2.8 sec. An in-house spectral-ion library file (.group) previously generated for human blood plasma proteins by searching.wiff format files generated in DDA mode against UniProtKB human FASTA database (UP000005640, 74,255 entries) using Proteinpilot™ Software 5.0 (SCIEX). A 1% global FDR at peptide level and 5% global FDR after excluding shared peptides (i.e. only unique peptides were included) at the protein level was considered for import in SWATH 2.0 microapp of PeakView 2.2 software (SCIEX). SWATH run files were added and retention time alignment was performed using peptides from abundant proteins. The processing settings for peak alignment were: maximum of 10 peptides per protein, 5 transitions per peptide, >95% peptide confidence threshold, 1% FDR. XIC extraction window was set to 55 min with 75 ppm XIC Width. All information was exported in the form of MarkerView (.mrkw) files. In MarkerView 1.2.1 (SCIEX), data normalization was performed using total area sum normalization and exported to excel.

**Co-occurrence analyses**. Co-occurrence among each pair of cytokines was calculated using Spearman correlation (r) and corresponding $p$-value of the correlation was measured using a t-distribution. Absolute values of the cytokines were used for the calculation of correlation network and threshold was set to r ≥ 0.7, $p < 0.01$ for the complete set of cytokines from SOC ($n = 40$) and CPT ($n = 39$) groups. All calculations were done using the 'Hmisc' R package and finally converted to a network file using the 'igraph' R package. Visualization of the network was performed using Cytoscape 3.7.2. Each cytokine was colour-coded and node size was set proportional to the fold change of median as compared to the same cytokine in the mild datasets.

**Statistical analyses and multivariate regression analysis**. All statistical analyses, as depicted in the results as well in appropriate figure legends, were performed using R and in some cases using Graphpad Prism 8 or STatistica64 (StatSoft). Primary outcomes of survival and recovery (in terms of discharge from hospital) were compared between the two arms using Kaplan–Meier Curve analysis —Mantel-Haenszel Hazard Ratio was calculated and statistical significance was tested by Mantel-Cox log-rank test.

**Reporting summary**. Further information on research design is available in the Nature Research Reporting Summary linked to this article.

## Data availability

All information regarding the availability of data and materials can be addressed to the corresponding authors. De-identified clinical data and experimental data are available on request sharing, which may need approval of the institutional ethical committees. The clinical outcome data for individual participants are provided in supplemental Table 4, which can be used for meta-analyses. The trial protocol is available as supplementary note 1 within the supplementary information file. The mass spectrometry proteomics data have been deposited to the ProteomeXchange Consortium via the PRIDE partner repository[46], with the dataset identifier PXD025453. The SARS-CoV-2 genome sequences have been uploaded on NCBI GenBank (https://www.ncbi.nlm.nih.gov/) with the GenBank accession number(s) OM169294-OM169315 and GISAID (https://www.gisaid.org/) with IDs between EPI_ISL_1672634–EPI_ISL_1672658. Source data are provided with this paper.

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

## Acknowledgements

We express their gratitude to Abhijit Chowdhury and Anurag Agrawal for their valuable guidance while conceiving the trial and Mohd. Faruq for help with RT-PCR experiments. D.G. acknowledges funding for the RCT and associated immune monitoring studies from Council of Scientific Industrial Research (CSIR), Govt. of India (MLP-129); R.P. acknowledges funding from CSIR (MLP-2005), Fondation Botnar (CLP-0031) and IUSSTF (CLP-0033).

## Author contributions

D.G. conceptualized, administered and supervised the study. D.G. and Y.R. designed the study protocol. D.G., S.C., P.B., R.D. and J.S. performed experiments. D.G., S.P., P.B., R.D., D.R.C., A.L., D.B. and R.B. analyzed the data. Y.R. and S.R.P. recruited patients on communication of the digitally generated randomization sequence from S.P., maintained clinical data and supervised clinical management. R.R., R.M., K.C., S.B., Ay.M., M.M.P., A.T., Av.M., S.M., A.R., M.S., B.S.S., A.H. and B.S. contributed to patient management. J.S.V., R.M., A.K. and R.P. did RT-PCR for SARS-CoV-2 and viral genome sequencing. P.S. and S.S. did the proteomics experiments. S.P. and P.B. recruited convalescent donors, D.B., C.M. and P.B. designed the donor selection guidelines, performed donor screening, apheresis and biobanking of convalescent plasma. D.G. wrote the manuscript with inputs from other authors. All authors approved the manuscript.

## Competing interests

The authors declare no competing interests.

## Additional information

[1]Infectious Disease & Beleghata General Hospital, Kolkata, India. [2]Department of Tropical Medicine, School of Tropical Medicine, Kolkata, India. [3]IICB-Translational Research Unit of Excellence, CSIR-Indian Institute of Chemical Biology, Kolkata, India. [4]Academy of Scientific and Innovative Research, Ghaziabad, India. [5]Division of Structural Biology & Bioinformatics, CSIR-Indian Institute of Chemical Biology, Kolkata, India. [6]CSIR-Institute of Genomics and Integrative Biology, Delhi, India. [7]Department of Pediatrics, Sagar Dutta Hospital & College of Medicine, Kolkata, India. [8]Department of Immunohematology & Blood Transfusion, Medical College, Kolkata, India. [9]Indian Statistical Institute, Kolkata, India. [10]Department of Critical Care Medicine, Tata Medical Center, Kolkata, India. ✉email: jaggs.nbmc@gmail.com; Dipyaman@iicb.res.in

