## [Peer Review File · Nature Communications]

Reviewers' Comments:

Reviewer #1:

Remarks to the Author:

The work presented by Ray et al focuses on an important and urgent issue to be studied and disseminated as a covid-19 treatment or significant survival benefit in severe COVID-19 patients with ARDS aged less than 67 years receiving CPT.

The manuscript is well written, and it is easy to follow.

I have mainly minor issues and some additional information requests as supplementary information and data upload.

Results

-The sentences should not start with a number in the recruitment section.

-Figure 1C should be better introduced. It is only clear after reading the text for figure 1D and for an expert in mass spectrometry.

-This reviewer is not familiar with all clinical terms, but whenever referring to oxygen (O₂) the "2" should always be subscript

Double-check these small typos:

-Line 435: "platelet count > 1,50,000" are the commas correct?

-Line 594 "dithiothreitol (Sigma-Aldrich, USA) and incubated at 56 °C for 25 minutes.", change ° for the degree symbol °

Same in

-Lane 597 digestion at an enzyme to substrate ratio of 1:10 for 18 hours at 37 °C.

-Lane 604 minutes for 10 min. Peptides were then separated on a reverse-phase C18 analytical column

-There are spaces missing between "for 10 min" and "C18 analytical"

-The MS data acquisition software is missing (Analyst? Which version?)

-How was the protein identification performed? Protein Pilot? Which version, database, and search parameters?

-What was considered a true protein identification? FDR analysis?

-The supplementary Table 1 is very limited on the information. Please provide the FDR excel file generated by Protein Pilot with the FDR analysis

-Raw data should also be placed in a public repository as PRIDE, and the code of that repository should be mentioned in the manuscript for further community data validation.

All graphs are a visual representation of data, but it is the data that is the real information. Please provide all data used to generate the graphs (supplementary tables). The study can only be used in a meta-analysis if the data can be fully retrieved and compared to other studies.

Reviewer #2:

Remarks to the Author:

The manuscript by Ray et al analyzes the effects of convalescent plasma in the treatment of severe COVID-19 patients through the means of a monocentric RCT. Despite the absence of clinical benefit in the overall study population – which is a result in line with previous RCTs (PMID: 33232588) – they interestingly found a significant beneficial effect in younger patients, as opposed to what has been previously described in a milder clinical setting (PMID: 33406353). The Authors also performed a thorough bioinformatic analysis and found in the anti-inflammatory proteomic features of CPT a possible explanation to the clinical benefit of this therapeutic approach. Some issues however need be addressed:

- Authors should better characterize the 2 study populations, including detailed information about other major prognostic determinants in COVID19 (age and comorbidities among others)

- Greater details should be also given to the treatments used that shows a known effect on COVID19 (such as the doses of corticosteroids in the two different groups)

- It should be analyzed if the aforementioned potential confounders were associated to different patient outcomes among the groups.

- The Authors affirm that a patient died before receiving the second dose of CPT and that death

was not related to the infusion; however, no information about the cause of death is given.

Reviewer #3:

Remarks to the Author:

This paper of Ray et al. present the results of a single center open label randomised controlled trial comparing treatment with convalescent plasma to standard of care in COVID-19 patients with medium to severe ARDS. I have reviewed the statistical aspects of this paper.

The authors motivate their study and the potential beneficial effect of the CP treatment well. This is a relatively small study (n=80), but fact that the treatment was allocated through a randomised procedure and most outcome measures are objectively measured makes the data valuable. I have two main concerns with this study.

1. It seems that the researchers did not follow any prespecified analysis plan.

a. The trial registration and the trial protocol that was submitted along with this paper lists 5 outcome parameters (in different order): 1. Progression to severe ARDS and recovery from severe ARDS 2. Immune correlates of response to plasma therapy 3. 'All cause' mortality 4. Time to become PCR negative from initial presentation 5. Incidence of minor and major transfusion reactions. It appears to me that not all of these are reported in the manuscript. Also the priority (primary outcome, secondary outcomes etc) is not remained for the ones that are analysed. The manuscript presents results of many other outcomes than these.

b. The sample size calculation totally unclear: for which outcome was it performed (it lists 'the incidence of symptomatic disease')? Why use a 'proportional odds model' when only mentioning 2 percentages. Such a proportional odds analysis was not mentioned in the manuscript.

2. The subgroup analyses using the cutoff <67 years versus ≥ 67 years appears to have been chosen in a highly data driven manner (figures 2A and 2B). No planned subgroup analysis was mentioned in the protocol nor in the trial registration. It is not clear how many other subgrouping options have been considered (based on other patient characteristics). Given the data-driven way of finding this subgroup in combination with the small sample size, there is a high risk of false positive findings. This subgroup finding should be placed in the light of all other subgroup analyses that have been performed and should be reported in a very exploratory sense instead of as the main result of the trial.

Minor points:

- No statistical paragraph for the clinical outcomes
- Unclear how the outcome time to discharge and was analysed – was death treated as a competing risk in these analyses? Also not clear how death was dealt with in the analysis of other endpoints such as S/F ratio.
- Some references to figure 3 are accidentally written as figure 4 in the text
- Some of the correlations can better be evaluated with Spearman than with Pearson correlation coefficient

Reviewer #4:

Remarks to the Author:

I was asked to comment on the viral genome sequencing and phylogenetics aspects of this study. These methods are used to compare genetic characteristics of the viruses between the two arms of the trial. They serve this purpose well, and the authors correctly interpret their results, stating that there are no clear-cut differences in viral genetic compositions between the two arms. Sequencing, base calling and phylogenetic approaches are standard, and I have no major concerns about them. In general, I expect the broad conclusions made by the authors (lack of differences in viral composition between the two arms) to be robust to method details. I have relatively minor comments.

1. According to Fig. 2D, many of the samples had large Ct values, up to 40. Was sequencing attempted for all samples, or did you use any cut-off for Ct values? From our experience, sequences from samples with Ct>30 are unreliable.

2. Please clarify in the Fig. 2B caption that the three colors for the branches indicate the three

clades (19A, 20A and 20B); this was not immediately obvious to me.

3. Please point out that the clade nomenclature you use is that of Nextstrain, and provide an appropriate reference.

4. It seems strange that clade 20B includes just two samples, although this clade has reached high frequencies in India early in the pandemic (70% by the end of the study (October), according to Nextstrain; <https://nextstrain.org/community/banijolly/Phylomis/COVID-India>). The authors might want to double-check the sequencing reads to make sure they have not miscalled the reference variant for the 'G28881A G28882A G28883C' triplet which characterizes 20B (https://nextstrain.github.io/ncov/naming_clades.html).

5. I may be missing it but are the sequences obtained deposited to a repository? I couldn't find the corresponding accession numbers.

Georgii Bazykin
(I sign all reviews)

Authors' responses to reviewers' comments on manuscript no. NCOMMS-21-06871-T

	Reviewers' comments	Authors' responses
Reviewer 1	The work presented by Ray et al focuses on an important and urgent issue to be studied and disseminated as a covid-19 treatment or significant survival benefit in severe COVID-19 patients with ARDS aged less than 67 years receiving CPT. The manuscript is well written, and it is easy to follow. I have mainly minor issues and some additional information requests as supplementary information and data upload.	The authors are thankful to the reviewer for appreciating the study, favourable assessment and inputs for further improvement of the manuscript.
	The sentences should not start with a number in the recruitment section.	We have corrected this in the revised manuscript.
	Figure 1C should be better introduced. It is only clear after reading the text for figure 1D and for an expert in mass spectrometry.	We have rearranged the discussion on these two panels of Figure 1 for easy readability and comprehension.
	-This reviewer is not familiar with all clinical terms, but whenever referring to oxygen (O₂) the "2" should always be subscript	We have corrected this inadvertent mistake in the revised manuscript.
	Double-check these small typos: -Line 435: "platelet count > 1,50,000" are the commas correct? -Line 594 "dithiothreitol (Sigma-Aldrich, USA) and incubated at 56 °C for 25 minutes.", change ° for the degree symbol ° Same in -Lane 597 digestion at an enzyme to substrate ratio of 1:10 for 18 hours at 37 °C. -Lane 604 minutes for 10 min. Peptides were then separated on a reverse-phase C18 analytical column -There are spaces missing between "for 10 min" and "C18 analytical"	We have corrected these inadvertent mistakes in the revised manuscript.
	The MS data acquisition software is missing (Analyst? Which version?)	MS data was acquired using Analyst TF Software v1.7.1 (SCIEX). This information has been now added in the revised manuscript.
	How was the protein identification	The spectral ion library used in this experiment

	performed? ProteinPilot? Which version, database, and search parameters?	was previously generated in lab for human blood plasma proteins by fractionating tryptic peptides using strong cation exchange (SCX) chromatography in 8 fractions. Each fraction was cleaned up using C18 Ziptips (Merck) as per manufactures protocol. LC-MS/MS data acquisition for each fraction was performed in DDA mode in a 87 min run using optimized source parameters and performing a full scan MS spectrum (400–1250 m/z) with an accumulated time of 0.25 second, followed by 30 MS/MS experiments (100–1500 m/z) with 0.05 second accumulation time each, on MS precursors with charge state 2+ to 5+ exceeding a 150 cps threshold. Rolling collision energy was used and the former target ions were excluded for 15 seconds. A combined database search for all the 8.wiff format DDA mode LC-MS/MS acquisition files was performed against UniProtKB human FASTA database (UP000005640, 74,255 entries) using Proteinpilot™ Software 5.0 (SCIEX) using Paragon algorithm to get protein identities. The parameters were set as follows: sample type- identification, cysteine alkylation—iodoacetamide, digestion—trypsin. Biological modification was enabled in ID focus. The search effort was set to ‘thorough ID’ and false discovery rate (FDR) analysis was enabled.
	What was considered a true protein identification? FDR analysis?	The spectral ion library was generated using data dependant acquisition (DDA). A 1% global FDR at peptide level and 5% global FDR after excluding shared peptides (i.e. only unique peptides were included) at protein level was considered for import in SWATH 2.0 microapp of PeakView 2.2 software (SCIEX). Subsequently, for SWATH analysis proteins were identified from the library using 1% peptide FDR. The information has now put in the methods section of the revised manuscript.
	The supplementary Table 1 is very limited on the information. Please provide the FDR excel file generated by Protein Pilot with the FDR analysis	FDR excel sheet has been uploaded on PRIDE database.
	Raw data should also be placed in a public repository as PRIDE, and the	The mass spectrometry proteomics data have been deposited to the ProteomeXchange

	code of that repository should be mentioned in the manuscript for further community data validation.	Consortium via the PRIDE partner repository with the dataset identifier PXD025453.
	All graphs are a visual representation of data, but it is the data that is the real information. Please provide all data used to generate the graphs (supplementary tables). The study can only be used in a meta-analysis if the data can be fully retrieved and compared to other studies.	
Reviewer 2	The manuscript by Ray et al analyzes the effects of convalescent plasma in the treatment of severe COVID-19 patients through the means of a monocentric RCT. Despite the absence of clinical benefit in the overall study population – which is a result in line with previous RCTs (PMID: 33232588) – they interestingly found a significant beneficial effect in younger patients, as opposed to what has been previously described in a milder clinical setting (PMID: 33406353). The Authors also performed a thorough biomolecular analysis and found in the anti-inflammatory proteomic features of CPT a possible explanation to the clinical benefit of this therapeutic approach. Some issues however need be addressed.	The authors thank the reviewer for appreciating the study, favourable assessment and inputs for further improvement of the manuscript.
	Authors should better characterize the 2 study populations, including detailed information about other major prognostic determinants in COVID19 (age and comorbidities among others)	The clinical and laboratory parameters are compared between the two arms in Table 1 in original manuscript, now supplemental table 2 in the revised manuscript. The distribution and co-occurrences of co-morbid conditions are included as venn diagrams in Figure 2E. The randomisation of the patients ensured a very comparable distribution of patients into the two arms. Subclass analyses for the two major co-morbidities (type 2 diabetes and hypertension) have now been included in figure 4I.
	Greater details should be also given to the treatments used that shows a known effect on COVID19	The composition of the standard of care therapy and the frequency of patients receiving specific pharmacotherapies are

	(such as the doses of corticosteroids in the two different groups)	included in supplemental table 2. An extended subclass analysis table has also been included as figure 4I.
	It should be analyzed if the aforementioned potential confounders were associated to different patient outcomes among the groups.	The randomisation of the patients ensured a very comparable distribution of patients into the two arms, as evident from the comparison of clinical characteristics provided in the manuscript. Still we also share the curiosity expressed here by the reviewer. But we refrained from more granular subclass analyses due to the small size of our cohort. Further classification of the patients would have failed to attain the statistically valid sample sizes for such analyses and as a result any conclusion drawn could have been misleading. In the revised manuscript we have now a few major sub-class analyses as Figure 4I, looking at clinical outcomes (survival) compared between the two arms in males, females, patients with type 2 diabetes or hypertension and patients who received corticosteroids and remdesivir. As expected from the low sample numbers none of these registered any significant difference.
	The Authors affirm that a patient died before receiving the second dose of CPT and that death was not related to the infusion; however, no information about the cause of death is given.	In the hospital record, the physician's note depicted that the patient was found dead following a fall on the ground at 11.30AM on the day after he was transfused with the first bag of CP (which was done in the afternoon on the preceding day). The patient had recent history of similar falls in the available medical records, but as this was not an exclusion criterion he was randomised into our study. No autopsy was performed. But the time of death precluded any transfusion reaction due to CP. The authors are of the opinion that this detailed information is not of great relevance for the present manuscript.
Reviewer 3	This paper of Ray et al. present the results of a single center open label randomised controlled trial comparing treatment with convalescent plasma to standard of care in COVID-19 patients with medium to severe ARDS. I have reviewed the statistical aspects of this paper. The authors motivate their study	The authors thank the reviewer for appreciating the study, favourable assessment and inputs for further improvement of the manuscript.

	and the potential beneficial effect of the CP treatment well. This is a relatively small study (n=80), but fact that the treatment was allocated through a randomised procedure and most outcome measures are objectively measured makes the data valuable. I have two main concerns with this study.	
	1. It seems that the researchers did not follow any prespecified analysis plan. a. The trial registration and the trial protocol that was submitted along with this paper lists 5 outcome parameters (in different order): 1. Progression to severe ARDS and recovery from severe ARDS 2. Immune correlates of response to plasma therapy 3. 'All cause' mortality 4. Time to become PCR negative from initial presentation 5. Incidence of minor and major transfusion reactions. It appears to me that not all of these are reported in the manuscript. Also the priority (primary outcome, secondary outcomes etc) is not remained for the ones that are analysed. The manuscript presents results of many other outcomes than these.	We agree with the reviewer on this. The subclass analysis was not pre-specified, but as it could derive clinically relevant information we attempted for it as depicted in figure 3. As depicted in the protocol represented in the Clinical Trial Registry of India (CTRI/2020/05/025209) the primary outcomes of the trial were  1. To compare all-cause mortality 2. To identify the immune correlates for response to plasma therapy. And the secondary outcomes were,  1. To compare recovery from ARDS in both groups 2. To compare time taken to negative viral RNA PCR 3. Adverse reaction to plasma therapy The analysis depicted in the manuscript thus encompassed the primary outcomes (survival between two groups, plasma cytokine levels in response to CPT and effect of CPT on it) as well as two of the secondary outcomes, viz. recovery in terms of discharge from the hospital and documenting adverse effects of CPT (found to be none in this trial). Due to operational issues in our low-resource clinical trial site amidst the pandemic arterial blood gas measurements were not regular, preventing us from accurately assigning ARDS status of each patients on daily basis as per international classifications, and thus we documented SpO₂/FiO₂ ratio based on O₂ saturation in the capillary blood. One of the secondary outcomes, viz. time taken to be RT-PCR negative could not be documented (originally planned to be documented from the data available from the health records) for any patients as when the

		actual trial was started the directive from the Indian Council of Medical Research precluded use of repeated RT-PCR in hospitalized patients and advised discharge from hospital on remission.
	b. The sample size calculation totally unclear: for which outcome was it performed (it lists ‘the incidence of symptomatic disease’)? Why use a ‘proportional odds model’ when only mentioning 2 percentages. Such a proportional odds analysis was not mentioned in the manuscript.	We thank the reviewer for pointing out this. The sample size was performed based on the outcome of severe ARDS or death, the phrase ‘the incidence of symptomatic disease’ was a overlooked mistake in the CDSCO protocol. The sample size was calculated using the proportional odds model, taking incidence of the specified outcomes in the preceding one month in the chosen clinical trial site among patients resenting with mild and moderate ARDS and receiving the standard of care advised by Indian Council of Medical Research at that point of time. For the analysis of our data we did not perform proportional odds and the primary outcome considered was comparison of survival till 30days and we performed Kaplan Meier Curve analysis to compare between the two arms by calculating the Mantel-Haenszel Hazard Ratio and performing Mantel-Cox log rank test, as depicted in the manuscript.
	2. The subgroup analyses using the cutoff <67 years versus >=67 years appears to have been chosen in a highly data driven manner (figures 2A and 2B). No planned subgroup analysis was mentioned in the protocol nor in the trial registration. It is not clear how many other subgrouping options have been considered (based on other patient characteristics). Given the data-driven way of finding this subgroup in combination with the small sample size, there is a high risk of false positive findings. This subgroup finding should be placed in the light of all other subgroup analyses that have been performed and should be reported in a very exploratory sense instead of as the main result of the trial.	We have described in the manuscript how we had led to the age-based sub-class analyses of the clinical outcome data (revised Figure 4A& B). We completely agree that these were not pre-specified analyses and were exploratory in nature. Regarding using other parameters for similar subclass analyses, in the original manuscript we refrained from more granular subclass analyses due to the small size of our cohort. Further classification of the patients would have failed to attain the statistically valid sample sizes for such analyses and as a result any conclusion drawn could have been misleading. In the revised manuscript we have now added a few major sub-class analyses as Figure 4I, looking at the primary outcome (survival) compared between the two arms in males, females, patients with type 2 diabetes or hypertension and patients who received

	corticosteroids and remdesivir. As expected possibly due to the low sample numbers none of these registered any significant difference.
No statistical paragraph for the clinical outcomes	The statistics adopted for the comparison of the clinical outcomes were described in the results section itself. We had a brief description on statistical methods adopted in the methods section. We have enriched it with more information in the revised manuscript.
Unclear how the outcome time to discharge and was analysed – was death treated as a competing risk in these analyses? Also not clear how death was dealt with in the analysis of other endpoints such as S/F ratio.	All deaths were taken as inability to achieve hospital discharge till 30 days. This had been mentioned in the figure legend. For S/F ratio kinetics data was taken as missing on days following deaths. We have now included the raw data table on day-wise S/F ratio values for individual patients. Adopted method of data imputation for missing values in case of $SFR_{7d}AUC$ calculation is described in the original manuscript.
Some references to figure 3 are accidentally written as figure 4 in the text	The authors than the reviewer to point out to these inadvertent mistakes. These have been corrected in the revised manuscript.
Some of the correlations can better be evaluated with Spearman than with Pearson correlation coefficient	We thank the reviewer for pointing it out. We had performed Pearson correlation analysis for all data projecting a linear relationship in case of significant associations. Now we have changed the method adopted to Spearman for data that show monotonic distribution with no evidence for linearity, viz. Figure 1A, Figure 2C and Figure 2D. As expected these correlations were also found to be statistically significant.
Reviewer 4	
I was asked to comment on the viral genome sequencing and phylogenetics aspects of this study. These methods are used to compare genetic characteristics of the viruses between the two arms of the trial. They serve this purpose well, and the authors correctly interpret their results, stating that there are no clear-cut differences in viral genetic compositions between the two arms. Sequencing, base calling and phylogenetic approaches are	The authors thank the reviewer for her appreciation of the study, favourable assessment and inputs for further enriching the manuscript.

	standard, and I have no major concerns about them. In general, I expect the broad conclusions made by the authors (lack of differences in viral composition between the two arms) to be robust to method details. I have relatively minor comments.	
	1. According to Fig. 2D, many of the samples had large Ct values, up to 40. Was sequencing attempted for all samples, or did you use any cut-off for Ct values? From our experience, sequences from samples with Ct>30 are unreliable.	We thank the reviewer for the query in this regard. Based on the RT-PCR kit (Cat No. 11NCO10, SD Biosensor) used for the RT-PCR of the samples used in this study, we followed the manufacturer's recommendation. The kit suggested using the cut-off of Ct value 36 for the SARS-CoV-2 genes (RdRp and E gene) and the performance of the human positive control gene to declare a sample as SARS-CoV-2 positive. Sequencing was attempted for all samples within the said cut-off. Of the 52 sequenced samples, there were 17 samples with Ct>30, 5 samples with Ct>35 and rest 30 samples of Ct<30. In terms of sequencing performance, inclusive of genome coverage and the sequencing depth, we did not observe correlation with the Ct values. We found that samples even with high Ct value performed well upon sequencing whereas lower Ct value samples had sub-optimal sequencing output. During analysis, the read quality has been kept uniform for all the samples used in the study irrespective of the background Ct value. Outside this study, using Nanopore and ARTIC protocol, we have achieved success with higher Ct value samples.
	2. Please clarify in the Fig. 2B caption that the three colors for the branches indicate the three clades (19A, 20A and 20B); this was not immediately obvious to me.	Thank you for pointing this out. We have edited to include this in the revised manuscript. Line no. 758 "The three colors, blue, red and green represent Nextstrain Clade 19A, 20A and 20B respectively".
	3. Please point out that the clade nomenclature you use is that of Nextstrain, and provide an appropriate reference.	We thank the reviewer for the suggestion. We have now included the information towards this in the revised manuscript. We have used the Nextstrain for clade nomenclature in this manuscript and have added the reference in Line no. 563 which reads as "Clade Nomenclature of Nextstrain was used to visualize the phylogenetic tree" to

		the revised manuscript (Hadfield J, Megill C, Bell SM, Huddleston J, Potter B, Callender C, Sagulenko P, Bedford T, Neher RA. Nextstrain: real-time tracking of pathogen evolution. Bioinformatics. 34, 4121-4123 (2018). doi: 10.1093/bioinformatics/bty407).																																				
	4. It seems strange that clade 20B includes just two samples, although this clade has reached high frequencies in India early in the pandemic (70% by the end of the study (October), according to Nextstrain; https://nextstrain.org/community/banijolly/Phylovis/COVID-India). The authors might want to double-check the sequencing reads to make sure they have not miscalled the reference variant for the 'G28881A G28882A G28883C' triplet which characterizes 20B (https://nextstrain.github.io/ncov/naming_clades.html).	We thank the reviewer and take this opportunity to share the below. The SARS-CoV-2 positive samples included in this manuscript were collected from the Eastern region of India, from Kolkata in the state of West Bengal. To double check the relatively lower percentage of 20B clades, we analyzed the SARS-CoV-2 genome sequences submitted in GISAID from this region of India. We looked for the GISAID deposited sequences from Kolkata as well as West Bengal (overall) and found 331 and 1065 genome sequences, respectively. The percentage of 20A and 20B clade is shown below, which shows the 20A clade is present in higher frequencies compared to 20B. Our analysis from all the available sequences (331 samples) in GISAID from the region of study (Kolkata) till date (10th April, 2021), 60.42% of these sequences are of Nextstrain clade 20A, while clade 20B is observed at a frequency of 26.59%. The lower percentage of clade 20B is also reflected when we analyze the total GISAID data from the state of West Bengal. GISAID Kolkata (331 Sequences)    Clade Number of sequences Percentage (out of 331)      19A 3 0.91    19B 16 4.83    20A 200 60.42    20B 88 26.59    20I/501Y.V 1 22 6.65    20H/501Y.V 2 2 0.60     GISAID West Bengal (1065 Sequences)    Clade Number of sequences Percentage (out of 1065)      19A 34 3.19    	Clade	Number of sequences	Percentage (out of 331)		19A	3	0.91		19B	16	4.83		20A	200	60.42		20B	88	26.59		20I/501Y.V 1	22	6.65		20H/501Y.V 2	2	0.60		Clade	Number of sequences	Percentage (out of 1065)		19A	34	3.19	
Clade	Number of sequences	Percentage (out of 331)																																				
19A	3	0.91																																				
19B	16	4.83																																				
20A	200	60.42																																				
20B	88	26.59																																				
20I/501Y.V 1	22	6.65																																				
20H/501Y.V 2	2	0.60																																				
Clade	Number of sequences	Percentage (out of 1065)																																				
19A	34	3.19																																				

		19B	21	1.97															
		20A	603	56.62															
		20B	357	33.52															
		20I/501Y.V1	42	3.94															
		20H/501Y.V	8	0.75															
		2																	
		We also specifically looked for the month of September-October 2020 (21 in number) from Kolkata, which corroborates the same pattern. GISAID Kolkata (September-October, 21 samples)    Clade Number of sequences Percentage (out of 21)     20A 16 76.19   20B 4 19.05   20I/501Y.V1 1 4.76    We also double-checked for the basecalls at positions G28881A, G28882A and G28883C and did not find any anomaly.						Clade	Number of sequences	Percentage (out of 21)	20A	16	76.19	20B	4	19.05	20I/501Y.V1	1	4.76
Clade	Number of sequences	Percentage (out of 21)																	
20A	16	76.19																	
20B	4	19.05																	
20I/501Y.V1	1	4.76																	
	5. I may be missing it but are the sequences obtained deposited to a repository? I couldn't find the corresponding accession numbers.	We thank the reviewer for the suggestion. We would submit the sequences to GISAID. Sequences with more than 50% coverage were uploaded.																	

Reviewers' Comments:

Reviewer #1:

Remarks to the Author:

Dear Authors

thanks for addressing almost all points I have raised.

There are just two points from my side.

Due to informatic issues and the Front being used on the communication I believe there was a miss understanding in the degrees Celsius issue. The proper form is very similar to what you presented, the number, for instance "56", then the degrees Celsius symbol, which is a small round superscript circle. What you had on your original manuscript was "56^o" and this symbol has the small superscript underlined. It is a small detail but important for data mining.

So please remove the "degrees Celsius" wording (probably the editorial team will solve this). Let's see if this time you receive what I want to write in the platform, it should be "56°C" and not "56^oC".

The second aspect was not addressed from my first round of comments:

"All graphs are a visual representation of data, but it is the data that is the real information. Please provide all data used to generate the graphs (supplementary tables). The study can only be used in a meta-analysis if the data can be fully retrieved and compared to other studies."

You don't need to prepare a PDF file with all this information. If you have it organized in excel, please provide such files. Again this is important for meta-analysis to cite your work and for data mining.

Reviewer #2:

Remarks to the Author:

The concerns of this reviewer have been addressed.

Reviewer #3:

Remarks to the Author:

Authors have given further clarifications regarding the statistical methodology that they have followed. They did not adjust main reporting of results. This has unfortunately only further increased my concerns with the manuscript. Methods used do not support the results as currently reported.

My biggest concern is about the age based subclass analysis (comparison between CP and SOC for the subgroup <67). A post-hoc highly granular data driven selective procedure was used to arrive at this cut-off, with >30 cut-offs tested. With that many tries it is not surprising that one of the cut-offs turns out (borderline) significant when using methods that do not account for multiple testing. Any unadjusted p-value based on this cut-off will be incorrect. Below I suggest some adjustments to the manuscript, but imo the full manuscript needs to be checked in order to overcome this concern.

Abstract:

Only report results here following from the pre-planned analyses. Do not mention any results of post-hoc exploratory subgroup analyses.

Introduction:

- Page 5 authors write ' and has been concluded on meeting the primary outcomes'. Not clear to me what this means. To avoid confusion, I suggest to omit this phrase.
- Do not refer to 'time to recovery (time till the day of discharge from hospital)' as a prespecified end-point as you have explained that it was not prespecified in this way.
- Whenever referring to results from the age-subgroup analysis, please place it in context of all other subgroup analyses that were performed and indicate that these were post-hoc exploratory subgroup analyses where the age cut-off was chosen by a data-greedy algorithm. Given the nature of these analyses, the p-values reported are not correct. Next to having tested at least 6 other subgroupings (figure 4I) and >30 age cut-offs analyses (figure 4B) and multiple outcomes, all without correcting for multiple testing, the cut-point for age was determined in the same dataset

as where its effect was tested. This lead to further over-optimism in the inference, i.e., the reported p-values will be much too low. So please remove any reference to finding 'significant' differences with age grouping. E.g. remove formulations like
o "we identified a major heterogeneity of response based on age...."

Results

- Do not refer to the age based subgroup analyses as being 'significant' (see above). This holds for all tested outcomes, not only survival. (the over-optimism from the selection procedure will carry over to all outcomes that are correlated to survival)

Discussion

- Authors currently state that " The clinical outcome comparisons revealed a significant benefit registered in severe COVID19 patients, most of who had progressed to moderate acute respiratory syndrome, and were aged less than 67 years." I do not agree with this conclusion. I believe the data was not able to not indicate any significant differences in clinical outcomes (survival 10/40 vs 14/40 and discharge showing more or less the opposite of those numbers).

Besides this main concern, the points from the consort statement need to be checked for completeness. Some of the points that do not seem to be complete:

- 8b The type of randomisation is not reported (was it eg simple randomisation, block randomisation)?
- 9 how was concealment ensured? The author contributions list that " Y.R. and S.R.P. recruited patients on communication of the digitally generated randomization sequence from S.P." Which suggests allocation was not concealed, is that correct? If so this is a major limitation and should be explicitly stated in the manuscript.
- 24 Where the full trial protocol can be accessed: not clear to me where can be downloaded. The protocol send along with the manuscript is not dated. Please specify al changes to the protocol made after the start of recruitment of patients by their timing and reason.
- Sample size calculation: not clear to me if the stated 28% vs 5% was chosen upfront or later on.

I still think the data are highly valuable to publish. However, reported conclusions should be in line with them. I strongly advise to include a (clinical) biostatistician in the team to ensure this.

Reviewer #4:

Remarks to the Author:

I am completely satisfied with the author's responses.

Georgii Bazykin

Authors' responses to reviewers' comments on manuscript no. NCOMMS-21-06871-A

	Reviewers' comments	Authors' responses
Reviewer 1	thanks for addressing almost all points I have raised. There are just two points from my side. Due to informatic issues and the Front being used on the communication I believe there was a miss understanding in the degrees Celsius issue. The proper form is very similar to what you presented, the number, for instance "56", then the degrees Celsius symbol, which is a small round superscript circle. What you had on your original manuscript was "56^o" and this symbol has the small superscript underlined. It is a small detail but important for data mining. So please remove the "degrees Celsius" wording (probably the editorial team will solve this). Let's see if this time you receive what I want to write in the platform, it should be "56°C" and not "56^oC". The second aspect was not addressed from my first round of comments: "All graphs are a visual representation of data, but it is the data that is the real information. Please provide all data used to generate the graphs (supplementary tables). The study can only be used in a meta-analysis if the data can be fully retrieved and compared to other studies." You don't need to prepare a PDF file with all this information. If you have it organized in excel, please provide such files. Again this is important for meta-analysis to cite your work and for data mining.	Authors are happy that the reviewer was satisfied with the revised manuscript. As rightly pointed out by the reviewer, the wordings of 'degree Celsius' has now been replaced. As recommended by the reviewer we have now provided the clinical meta-data, comprising of age, gender, pre-existing co-morbidities and clinical outcome, as supplemental table 4, to enable efforts at meta-analyses. Authors will like point out here, already several meta-analysis have included data from this study for which the authors have provided all necessary data.
Reviewer 2	The concerns of this reviewer have been addressed.	Authors are happy that the reviewer was satisfied with the revised manuscript.
Reviewer 3	Authors have given further clarifications regarding the statistical methodology that they have followed. They did not adjust main reporting of results. This has unfortunately only	Authors are happy that the reviewer appreciated greater clarifications provided in the revised manuscript. As suggested by the reviewer, as well as the editor, now we also have adjusted the main

	further increased my concerns with the manuscript. Methods used do not support the results as currently reported.	reporting of results, focussing only on the prespecified outcomes. Mention of all exploratory analyses have been removed from abstract and main conclusions. Wherever the data from exploratory analyses are discussed they are duly flagged as not being prespecified.
	My biggest concern is about the age based subclass analysis (comparison between CP and SOC for the subgroup <67). A post-hoc highly granular data driven selective procedure was used to arrive at this cut-off, with >30 cut-offs tested. With that many tries it is not surprising that one of the cut-offs turns out (borderline) significant when using methods that do not account for multiple testing. Any unadjusted p-value based on this cut-off will be incorrect. Below I suggest some adjustments to the manuscript, but into the full manuscript needs to be checked in order to overcome this concern.	As suggested by the reviewer, as well as by the editor, in the re-revised manuscript we have specified how the trial dealt with all the primary and secondary outcomes with due diligence. As suggested we have also repeatedly flagged the non-pre-specified secondary analyses as and when they were described. The exploratory age-based subclass outcome data has now been put in supplemental materials as supplemental figure 4.
	Abstract: Only report results here following from the pre-planned analyses. Do not mention any results of post-hoc exploratory subgroup analyses. Introduction:	The title and the abstract of the manuscript have also been revised to omit data from all the non-prespecified exploratory sub-class analyses.
	- Page 5 authors write ' and has been concluded on meeting the primary outcomes'. Not clear to me what this means. To avoid confusion, I suggest to omit this phrase.	We have omitted this as suggested.
	- Do not refer to 'time to recovery (time till the day of discharge from hospital)' as a prespecified end-point as you have explained that it was not prespecified in this way.	We have now mentioned that 'the time to recovery' was not a prespecified outcome of the trial.
	- Whenever referring to results from the age-subgroup analysis, please place it in context of all other subgroup analyses that were performed and indicate that these were post-hoc exploratory subgroup	We have now put the age-based subclass analyses as supplemental figure 4 and all mention on this are flagged to be non-prespecified.

	analyses where the age cut-off was chosen by a data-greedy algorithm. Given the nature of these analyses, the p-values reported are not correct. Next to having tested at least 6 other subgroupings (figure 4I) and >30 age cut-offs analyses (figure 4B) and multiple outcomes, all without correcting for multiple testing, the cut-point for age was determined in the same dataset as where its effect was tested. This lead to further over-optimism in the inference, i.e., the reported p-values will be much too low. So please remove any reference to finding ‘significant’ differences with age grouping. E.g. remove formulations like  o “we identified a major heterogeneity of response based on age...” 	
	Results  - Do not refer to the age based subgroup analyses as being ‘significant’ (see above). This holds for all tested outcomes, not only survival. (the over-optimism from the selection procedure will carry over to all outcomes that are correlated to survival) 	We have now put the age-based subclass analyses as supplemental figure 4 and all mention on this are flagged to be non-prespecified. The exploratory analyses are now described solely in relation to immunological parameters.
	Discussion  - Authors currently state that “ The clinical outcome comparisons revealed a significant benefit registered in severe COVID19 patients, most of who had progressed to moderate acute respiratory syndrome, and were aged less than 67 years.” I do not agree with this conclusion. I believe the data was not able to not indicate any significant differences in clinical outcomes (survival 10/40 vs 14/40 and discharge showing more or less the opposite of those numbers). 	Now the discussion clearly mentions that based on the prespecified analyses the RCT showed no relative benefit on CPT in sever COVID-19 patients. All discussions on exploratory analyses are flagged.
	Besides this main concern, the points from the consort statement need to be checked for completeness. Some of the points that do not seem to be complete:	We thank the reviewer for pointing out this omission. The nature of the random sequence is now provided (four blocks of twenty).

	- 8b The type of randomisation is not reported (was it eg simple randomisation, block randomisation)?	
	- 9 how was concealment ensured? The author contributions list that “ Y.R. and S.R.P. recruited patients on communication of the digitally generated randomization sequence from S.P.” Which suggests allocation was not concealed, is that correct? If so this is a major limitation and should be explicitly stated in the manuscript.	The patient recruitment based on inclusion criteria preceded the communication of randomised sequence. Thereafter the allocation was not concealed. It is now mentioned clearly in the manuscript.
	- 24 Where the full trial protocol can be accessed: not clear to me where can be downloaded. The protocol send along with the manuscript is not dated. Please specify al changes to the protocol made after the start of recruitment of patients by their timing and reason.	A scanned copy of the signed trial protocol has now been retrieved from the institutional repository and uploaded. The inability to gather data from the secondary outcome 2 was informed to the institutional IRB very soon after beginning the trial in June 2020. The change pertaining to secondary outcome 1 has also been communicated post-facto to the concerned IRB.
	- Sample size calculation: not clear to me if the stated 28% vs 5% was chosen upfront or later on.	The trial was designed to recruit 40 patients in each arm, based on sample size calculation based incidence of the specified primary outcome among patients presenting with mild and moderate ARDS and receiving standard of care (as per contemporaneous management guidelines of Indian Council of Medical Research as well as Dept. of Health, Govt. of West Bengal) for the preceding one month in the same clinical site (28% with standard of care, with hypothesized incidence of the same in response to CPT taken to be 5%).
Reviewer 4	I am completely satisfied with the author's responses.	Authors are happy that the reviewer was satisfied with the revised manuscript.